

# Bioinformatics analysis identifies several intrinsically disordered human E3 ubiquitin-protein ligases

Wouter Boomsma, Sofie V. Nielsen, Kresten Lindorff-Larsen, Rasmus Hartmann-Petersen and Lars Ellgaard

Linderstrøm-Lang Centre for Protein Science, Department of Biology, University of Copenhagen, Copenhagen, Denmark

## ABSTRACT

The ubiquitin-proteasome system targets misfolded proteins for degradation. Since the accumulation of such proteins is potentially harmful for the cell, their prompt removal is important. E3 ubiquitin-protein ligases mediate substrate ubiquitination by bringing together the substrate with an E2 ubiquitin-conjugating enzyme, which transfers ubiquitin to the substrate. For misfolded proteins, substrate recognition is generally delegated to molecular chaperones that subsequently interact with specific E3 ligases. An important exception is San1, a yeast E3 ligase. San1 harbors extensive regions of intrinsic disorder, which provide both conformational flexibility and sites for direct recognition of misfolded targets of vastly different conformations. So far, no mammalian ortholog of San1 is known, nor is it clear whether other E3 ligases utilize disordered regions for substrate recognition. Here, we conduct a bioinformatics analysis to examine >600 human and *S. cerevisiae* E3 ligases to identify enzymes that are similar to San1 in terms of function and/or mechanism of substrate recognition. An initial sequence-based database search was found to detect candidates primarily based on the homology of their ordered regions, and did not capture the unique disorder patterns that encode the functional mechanism of San1. However, by searching specifically for key features of the San1 sequence, such as long regions of intrinsic disorder embedded with short stretches predicted to be suitable for substrate interaction, we identified several E3 ligases with these characteristics. Our initial analysis revealed that another remarkable trait of San1 is shared with several candidate E3 ligases: long stretches of complete lysine suppression, which in San1 limits auto-ubiquitination. We encode these characteristic features into a San1 similarity-score, and present a set of proteins that are plausible candidates as San1 counterparts in humans. In conclusion, our work indicates that San1 is not a unique case, and that several other yeast and human E3 ligases have sequence properties that may allow them to recognize substrates by a similar mechanism as San1.

Corresponding authors
Wouter Boomsma, wb@bio.ku.dk
Lars Ellgaard, lellgaard@bio.ku.dk

Subjects Bioinformatics, Cell Biology, Biochemistry
Keywords E3 ligase, Hrd1, San1, Intrinsic disorder, STUbL, Ubiquitin, Protein folding, Protein degradation, Quality control

## INTRODUCTION

A coordinated and efficient regulation of protein levels is of crucial importance to all cells, and ubiquitin-mediated proteasomal degradation plays a central role in protein

homeostasis. In addition to its role in controlling protein levels for regulatory purposes, the proteasome also degrades misfolded proteins that may otherwise accumulate as toxic protein aggregates.

Cellular proteins are targeted for proteasomal degradation by the attachment of multiple moieties of the small regulatory protein ubiquitin, most often to lysine residues, but in some cases also to serines, threonines or cysteines (*Wang, Herr & Hansen, 2012*). This process occurs through the sequential action of a ubiquitin-activating enzyme (a so-called E1 enzyme), a ubiquitin-conjugating enzyme (E2) and a ubiquitin-protein ligase (E3). First, an E1 enzyme gets activated with a ubiquitin moiety in an ATP-dependent process, and subsequently passes on ubiquitin to an E2 enzyme. The E2 enzyme then interacts with an E3 ubiquitin-protein ligase. The most common class of E3 ligases contains a Really Interesting New Gene (RING) domain, which acts to bind ubiquitin-conjugating E2 enzymes (*Metzger et al., 2014*). RING E3 ligases thus bridge ubiquitin-charged E2s with substrates, which then become ubiquitinated. Multiple rounds of this process ensure substrate poly-ubiquitination, a requirement for degradation of most proteasomal substrates (for a recent review of the ubiquitin-proteasome system see *Kleiger & Mayor, 2014*).

Although most components and the basic overall mechanisms of this system are well understood, we still lack a comprehensive molecular understanding of how proteins are recognized and marked for degradation. In some cases, E3 enzymes bind ancillary protein factors that aid in the recognition of the substrate. The E3 enzyme CHIP, for example, ubiquitinates misfolded proteins that are recognized and delivered to CHIP via Hsp70 chaperones (*Arndt, Rogon & Hohfeld, 2007*). In this case, recognition of partially folded or misfolded substrates is offloaded to a chaperone, and the E3 enzyme then recognizes the substrate-loaded Hsp70.

The use of protein co-factors to aid in substrate recognition is one solution to the problem of how a fixed number of E3 enzymes can recognize the very large and conformationally diverse set of substrates that the system must be able to deal with. Recently, however, an alternative mechanism has been suggested for how the San1 E3 ligase from *Saccharomyces cerevisiae* can recognize misfolded substrates. San1, a protein of 610 amino acids with a RING domain spanning residues 165–280, appears to be largely devoid of well-defined three-dimensional structure (*Rosenbaum et al., 2011*). Although the cytosolic Hsp70 chaperone Ssa1 and the ring-shaped Cdc48/p97 chaperone can assist in degradation of San1 substrates (*Gallagher, Clowes Candadai & Gardner, 2014*; *Guerriero, Weiberth & Brodsky, 2013*; *Kriegenburg et al., 2014*), a number of biochemical experiments strongly suggest that the disordered parts of the San1 sequence provide direct binding sites for substrate proteins targeted for degradation (*Rosenbaum et al., 2011*). In particular, throughout the disordered regions of San1 there are several short hydrophobic stretches that function as substrate binding sites (*Rosenbaum et al., 2011*). This distribution of substrate binding patches, combined with the high flexibility of the disordered protein, allows San1 to interact with multiple different misfolded proteins and bypasses the need for chaperones. Hence, the intrinsic disorder present in San1 apparently underlies its ability to recognize misfolded proteins. However, potentially due to the poor sequence conservation of disordered proteins, it has not yet been possible to identify any San1 orthologs in higher eukaryotes.

Intrinsic disorder in proteins is now recognized to be broadly distributed across all kingdoms of life, and to play an important role in a large number of biological processes (*Dyson & Wright, 2005*). The large flexibility of fully intrinsically disordered proteins, or long stretches of intrinsic disorder in conjunction with well-folded domains, is believed to provide a number of particular properties to proteins (*Babu, Kriwacki & Pappu, 2012*). Importantly, the heterogeneous structures attained by these proteins allow them to bind a broad range of target proteins, either through a process called folding-upon-binding in which the disordered regions gain a specific structure upon binding to a target, or through the formation of "fuzzy complexes" in which substantial disorder is retained (*Cumberworth et al., 2013*).

The finding that San1 can recognize its protein substrates directly via its intrinsically disordered regions prompted us to explore whether this mechanism might be employed by other E3 enzymes. Using sequence conservation as a search criterion proved to be insufficient in this respect, presumably due to the low levels of sequence conservation that characterize intrinsically disordered protein regions. As an alternative, site-specific disorder profiles have been shown to display substantial preservation through evolution (*Brown, Johnson & Daughdrill, 2010*; *Brown et al., 2011*; *Chen et al., 2006a*; *Chen et al., 2006b*; *Daughdrill et al., 2007*), and have recently been used as the basis for detecting similar disorder functionalities across species (*Mahani, Henriksson & Wright, 2013*; *Petrovich et al., 2015*). We employ such an approach here, by constructing a procedure that searches for disorder properties similar to those of San1, combined with information about the unique pattern of binding regions known from the San1 protein. A previous study of structural disorder in the human ubiquitin-proteasome system found evidence for disorder in E1, E2 and—to the greatest extent—E3 enzymes, and provided examples for how disordered regions in E3 enzymes can function either as flexible linker regions or sometimes as sites for interaction with folded domains in substrates (*Bhowmick et al., 2013*). Our work extends this analysis by focusing on the extent to which E3 enzymes may use their intrinsic disorder in direct substrate recognition. The method was applied on datasets of 563 human and 80 yeast E3 enzymes. We demonstrate that a number of human E3 enzymes, in addition to displaying pervasive protein disorder, show two other hallmarks of the San1 sequence: short stretches of higher order potentially involved in substrate binding and a remarkably low content of lysine residues. Overall, our data indicate that San1 is not a unique case among E3 ligases and that a number of these enzymes could display a substrate recognition mode similar to San1.

## RESULTS

### Sequence-conservation based analyses reveal no obvious human orthologs of San1

As yet, no human ortholog of yeast San1 is known. In a first attempt to identify such a protein, we focused on the unusual RING domain of San1. Whereas a classical RING domain has a consensus motif of Cys-$X_2$-Cys-$X_{9-39}$-Cys-$X_{1-3}$-His-$X_{2-3}$-Cys/Asn/His-$X_2$-Cys-$X_{4-48}$-Cys-$X_2$-Cys (where X represents any amino acid residue) (*Budhidarmo,*
*Nakatani & Day, 2012*), the distance between the first Cys-X$_2$-Cys sequence and the following cysteine in the San1 RING domain is 88 residues (in *S. cerevisiae*), making it unusually long. However, using MOTIF search (http://www.genome.jp/tools/motif/), an algorithm designed to detect a specific sequence motif, we did not find obvious San1 candidates among the human proteins identified. Moreover, using the *S. cerevisiae* San1 RING domain in PSI-BLAST and DELTA-BLAST (*Altschul et al., 1997*) searches we did also not detect any human E3 ligase comprising the same characteristic RING domain as San1. To increase the sensitivity of the sequence-based search, we proceeded with an HMMSearch protocol, where the San1 sequence was encoded into a hidden Markov model using the jackhmmer program (both part of the HMMer software package; http://hmmer.org/). This procedure detected a number of potential matches, but the hits were predominantly defined by similarities in the ordered regions of San1—primarily the RING domain, and the detected homologues did not generally display the disorder and binding patterns characteristic of San1 (Fig. S1). In light of these results, we decided to search for human E3 ligases that contain characteristic sequence features of San1 other than those associated with the RING domain.

## Initial San1-similarity analysis

Although San1 from *S. cerevisiae* is largely intrinsically disordered outside of the RING domain, the flexible parts are predicted to contain short interspersed regions of higher order that coincide with sequence conservation among orthologs in other yeast species (*Rosenbaum et al., 2011*). Experimentally, it has been verified that these more ordered regions confer substrate binding (*Rosenbaum et al., 2011*). The overall model that emerges is therefore one where multiple binding sites (independently or in conjunction) can engage the substrates, whereas the flexibility provided by the intrinsic disorder allows San1 to adopt its structure to a wide variety of misfolded substrates (*Rosenbaum & Gardner, 2011*). We therefore analyzed datasets of *S. cerevisiae* and human E3s to identify E3 enzymes containing the same overall disorder/order signature as San1, in addition to a high overall content of intrinsic disorder. Specifically, we employed the consensus disorder annotation from the MobiDB database (*Potenza et al., 2015*) to identify regions of disorder in E3 enzymes (Fig. 1A, Table S1). For the San1 sequence, 78% was predicted to be disordered (Fig. 1B, Table S1).

We used the ANCHOR predictor (*Meszaros, Simon & Dosztanyi, 2009*), which has also been used in the characterization of San1 (*Rosenbaum et al., 2011*), to identify short regions within the disordered parts of the E3 enzymes with a potential for molecular interactions. Using only the amino acid sequence, ANCHOR predicts potential interaction sites present within disordered regions. When employing a 0.5 threshold level, 53% of all residues in San1 were predicted to be ANCHOR positive, with only four of the human E3 proteins scoring higher (Table S1). In the case of San1, the regions of higher order that coincide with sequence conservation among orthologs are generally 15–40 residues long (*Rosenbaum et al., 2011*). To create an output to better detect a disorder/order pattern similar to the one in San1, we therefore identified the total number of residues contained in ANCHOR positive

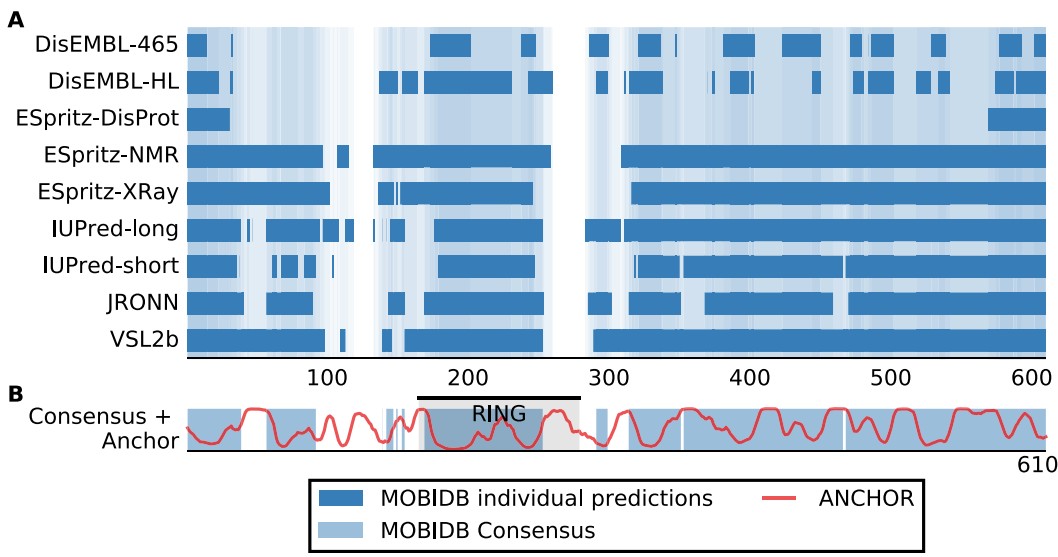

**Figure 1** **San1 is highly disordered and contains a characteristic pattern of interspersed ANCHOR positive regions for substrate interaction.** (A) Disorder annotation of the *S. cerevisiae* San1 protein, as provided by the MobiDB database (*Potenza et al., 2015*). Nine individual predictors were used by MobiDB as the basis for a consensus annotation, which we used for all analyses in this paper. The darkness of the shade of blue indicates how many of the nine predictors agree about the disorder annotation at a particular site. (B) Combined plot of the MobiDB consensus disorder annotation (blue) and the ANCHOR prediction (red) for San1. The position of the RING domain is indicated. This compact view is used as the basis for the comparative analyses throughout the paper.

regions of length 15–40 in each protein of the datasets, and normalized it to the length of the protein.

Plotting this ANCHOR score against the consensus disorder, and considering the candidates with highest scores in both dimensions, we indeed found San1 at the top of the yeast dataset, accompanied by a few other high-scoring candidates (e.g., Slx8; Figs. 2 and 3). The disorder and ANCHOR profiles for several of the top candidates in the human dataset (e.g., Rnf6, Rnf12, Pja1, Rn111; Figs. 2 and 3) showed clear similarities to the San1 profile, with general high levels of disorder and the distinct ANCHOR propensity pattern. This initial approach, based on a simple combination of disorder and anchor score, however, had a few shortcomings. First of all, it relies on global high values of disorder, thereby excluding proteins containing one or more ordered domains. Secondly, it fails to take into account another remarkable feature of San1: long segments without a single occurrence of lysine residues. Both these limitations are addressed below.

## San1-like E3s show substantial lysine suppression

San1 has previously been reported to contain unusually few lysine residues (*Fredrickson et al., 2013*) (Fig. 3 and Table S1). Since lysine is the most commonly used residue for ubiquitin conjugation, and based on experimental evidence that the introduction of lysines significantly destabilized the protein in cells, it was proposed that the low level of lysines in San1 protects the protein from auto-ubiquitination and therefore from degradation (*Fredrickson et al., 2013*). The few lysines present in San1 are found either within the RING

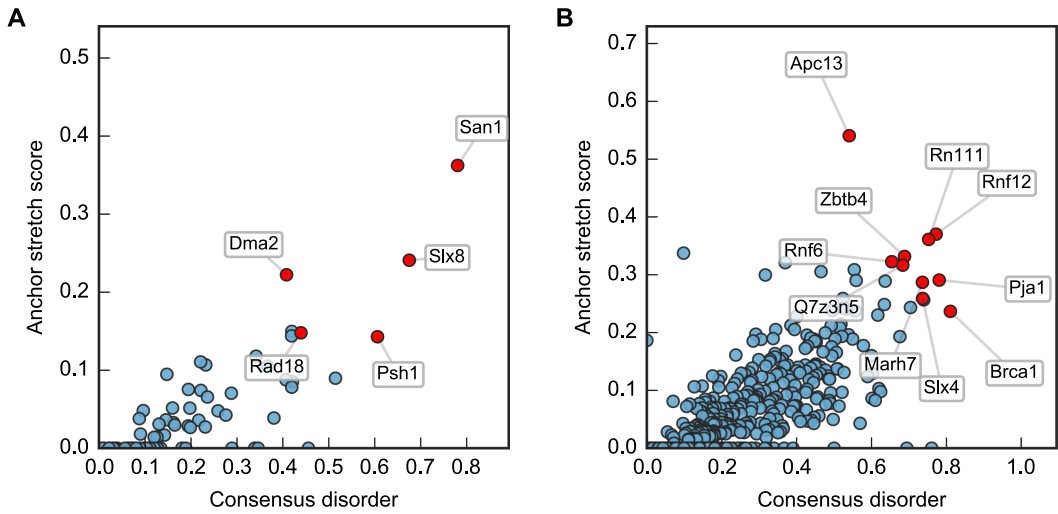

**Figure 2** **Characterization of the E3 dataset in terms of disorder and protein binding sites in disordered regions.** Plots of ANCHOR score versus fraction consensus disorder for all *S. cerevisiae* (A) and human (B) proteins in the datasets. Protein names are given for the top-scoring candidates, for which disorder and ANCHOR profiles are depicted in Fig. 3.

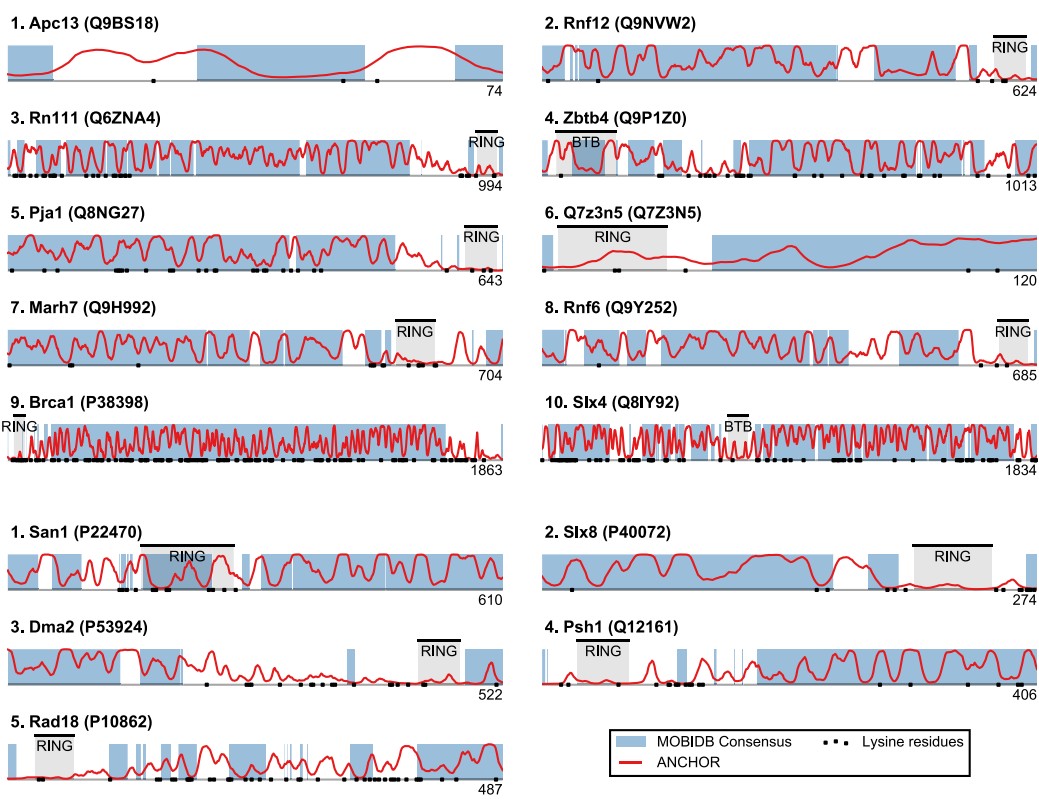

**Figure 3** **Identified San1-like E3 ligases show unusually low lysine content outside of the RING domain.** Plots of the highest-scoring candidates obtained using a score consisting of the product of disorder and anchor-stretch score. Each occurrence of a lysine residue is marked with a black dot.
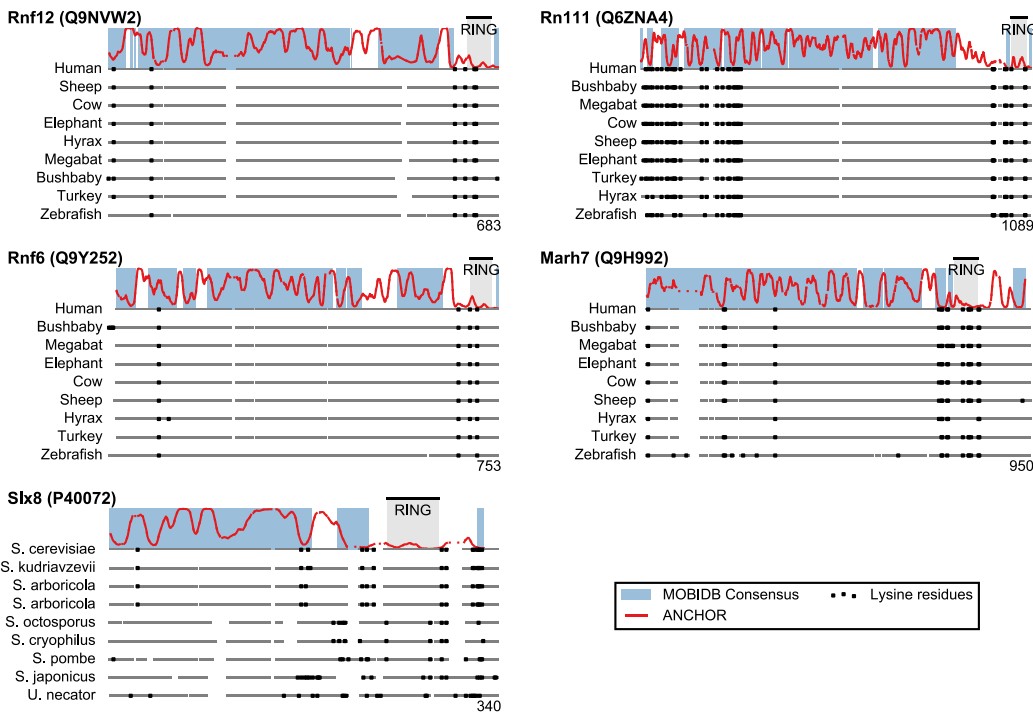

**Figure 4** **Low lysine content in San1 candidates is conserved broadly across species.** The lysine distributions of the candidates with strongest lysine suppression in Fig. 3 are compared to select orthologs from representative species of evolutionary divergence, suggesting lysine suppression to be a strongly conserved feature in these candidates. Note that the number below the alignment denotes the length of the alignment (including gaps), rather than the query sequence length.

domain or in a short sequence stretch of relatively low disorder immediately adjacent to this domain (Fig. 3, black dots), a feature which seems to be shared with several of the highest scoring candidates (Fig. 3), in some cases to a remarkable extent (e.g., >600 consecutive residues were devoid of lysines in the case of RN111). The most significant cases are highlighted in Fig. 4, demonstrating that the lysine suppression is strongly conserved among orthologs in a range of different species. The lack of lysines is particularly remarkable in light of the finding that disordered proteins are typically enriched in lysines compared to folded proteins (*Tompa, 2002*). Taken together, these results suggest that the San1-like E3 ligases identified here could employ the same strategy as San1 itself to avoid auto-ubiquitination and degradation (but see also 'Discussion').

## Definition of a San1 similarity score
Based on the considerations above, we developed a single score that quantifies whether a protein displays general sequence properties that are similar to those of San1. The score is based on disorder propensity, ANCHOR score and lysine suppression using a sliding window approach. For each protein, every residue is assigned a score based on the three quantities, calculated over a window centered at the residue in question (see Fig. 5): (a) the percentage of residues with consensus disorder larger than 0.5, (b) the percentage of residues within ANCHOR positive stretches, and (c) the longest lysine-free stretch of

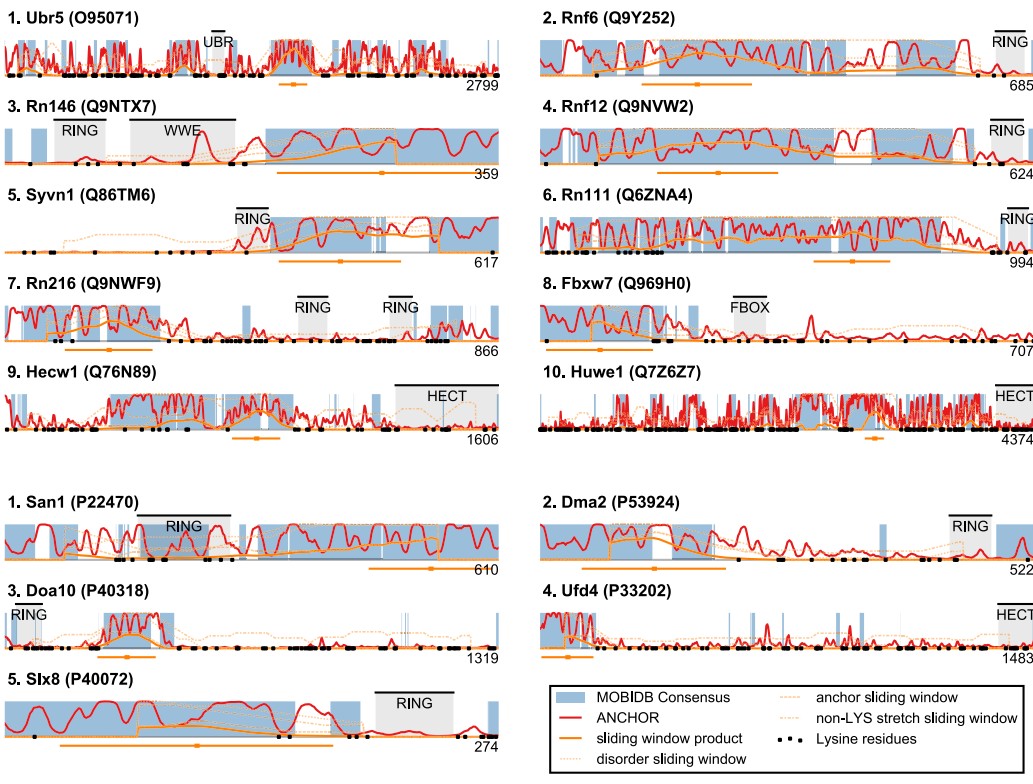

**Figure 5** **San1-like candidates can be detected using a feature-based similarity score.** Top-scoring candidates retrieved when applying the sliding-window San1-similarity score (see Table S1). Three complementary quantities are calculated over a window of 150 residues: (1) the percentage of disorder, (2) the percentage of residues within an ANCHOR positive stretch of length 15–40 and (3) the percentage of residues occurring in the maximum lysine-free stretch within the window. The San1-similarity score is the maximum value of the product of these numbers, with the corresponding window depicted as an orange line under each plot. The window size is chosen to be of fixed size, and the score is therefore not defined for the first and last 75 positions. Note that certain of the top-scoring candidates found using the San1 similarity score overlap with our initial, simpler approach that considered only disorder and ANCHOR score over the entire sequence (see Fig. 3).

residues, normalized by the length of the window (chosen here as 150 residues). For each position in the sequence, a product of these three quantities is calculated, and each protein is represented with the maximum value ("the San1 similarity score") observed among its residues. Thus, proteins with a high San1 similarity score contain a particular region that simultaneously has a high level of disorder, many potential sites for molecular interaction and strong lysine suppression.

The top-scoring candidates based on this approach are presented in Fig. 5 (the full list is available as Table S1). As expected, we find that San1 has one of the highest scores, with only a few human candidates scoring higher. In general, we recover many of the candidates found in the initial San1-similarity analysis presented above. As opposed to the initial analysis, the window-based approach allows the identification of larger proteins with a relatively low overall disorder, often reflecting the presence of one or more ordered regions (domains). Two such examples include human Syvn1/Hrd1 and yeast Dma2 (Fig. 5). For

a further discussion of the choice of parameters and how they affect the San1-similarity score we refer to the 'Methods' section.

## Analysis of potential functional orthologs of San1

Based on the bioinformatics investigation described above, we next analyzed specific proteins with high San1 similarity in more detail founded on the current literature. Specifically, among the top scoring hits in Fig. 5, we consider those candidates with known links to protein quality control, i.e., the human E3s Syvn1/Hrd1 and Hecw1 and the yeast E3s Doa10, Slx8, and Ufd4.

A well-characterized protein appearing in our analysis is human Syvn1/Hrd1, a multispanning transmembrane protein of the ER. The low overall disorder displayed by this protein is due to the transmembrane region ending at around residue 240, followed by a disordered C-terminal cytosolic region, which is also low in lysine content (Fig. 5). Curiously, the cytosolic tail of the budding yeast protein contains more lysines than in its orthologs. Whether this difference reflects a basic difference in the respective functional mechanisms is presently unknown.

Syvn1/Hrd1 is of particular interest in the current context, because it is known to interact with misfolded proteins. Thus, Syvn1/Hrd1 is involved in targeting misfolded ER and secretory proteins for proteasomal degradation in the cytosol through a process known as ER-associated degradation (ERAD) (*Christianson & Ye, 2014*). In this process, misfolded proteins are first recognized by ER lumenal chaperones and retrotranslocated across the ER membrane. When emerging in the cytosol, substrate proteins are ubiquitinated by ER-membrane localized E3 ligases, such as Syvn1/Hrd1, before they are finally degraded by the 26S proteasome (*Christianson & Ye, 2014*). Along with the RING domain, the predicted intrinsically disordered area in HRD1 faces the cytosol, and has so far not been ascribed any particular function. Presumably, Syvn1/Hrd1 transiently engages the at least partially extended ERAD substrates during ubiquitination, and the disordered region could therefore play a role, similar to the situation for San1, in this function.

Intriguingly, another ERAD E3, Doa10, was one of the best scoring yeast E3s. Like Syvn1/Hrd1, Doa10 is a multispanning transmembrane protein localized to the ER and inner nuclear membrane where it targets a variety of misfolded proteins, e.g., squalene monooxygenase, an enzyme involved in sterol homeostasis (*Foresti et al., 2013*), for proteasomal degradation (*Habeck et al., 2015* and references therein). As with Syvn1/Hrd1 the disordered region of Doa10 faces the cytosol, which fits well with its established role in targeting also soluble nuclear and cytosolic proteins for degradation (*Ravid, Kreft & Hochstrasser, 2006*).

Slx8 is a well-characterized SUMO-targeting ubiquitin ligase (STUbL). In the yeast nucleus, Slx8 forms a heterodimer with Slx5 (*Yang, Mullen & Brill, 2006*) and targets a number of sumoylated substrates for ubiquitin-dependent degradation (*Uzunova et al., 2007*). At least part of the disordered region of Slx8 is required for heterodimer formation with Slx5 (*Westerbeck et al., 2014*; *Yang, Mullen & Brill, 2006*), which contains the SUMO interacting motifs that are required for substrate recognition. Similar to San1, the Slx5-Slx8 heterodimer has been shown to participate in protein quality control. Both *SLX5* and

*SLX8* were identified as suppressors of a temperature sensitive mutant in the DNA binding protein Mot1 (*Wang, Jones & Prelich, 2006*). At high temperatures, the mutant Mot1 protein is highly unstable, but stabilized in *SLX5* and *SLX8* null mutants (*Wang, Jones & Prelich, 2006*). This is similar to how San1 was first connected to protein quality control of the mutant proteins Sir4-9 and Cdc68-1 (*Gardner, Nelson & Gottschling, 2005*), and suggests that the mutant Mot1 protein is still functional at the restrictive temperature, but targeted for degradation. However, the degradation of Mot1 was also shown to depend on the SUMO-ligases Siz1 and Siz2 (*Wang, Jones & Prelich, 2006*), revealing that sumoylation of Mot1 was required for Slx5-Slx8 dependent degradation. Although in sequence, they do not appear highly conserved, the mammalian STUbL Rnf4/Snurf is a functional ortholog of budding yeast Slx5-Slx8 (*Prudden et al., 2007*; *Sriramachandran & Dohmen, 2014*). However, as opposed to the situation with Slx5-Slx8 in yeast, Rnf4/Snurf forms a homodimer (*Plechanovova et al., 2011*), which has primarily been connected with the SUMO-dependent degradation of the Pml-Rarα oncoprotein (*Tatham et al., 2008*). Of note, Rnf4/Snurf did not display a high San1-similarity score. However, another less characterized human STUbL, Rn111 (Fig. 5), which is also involved in targeting Pml-Rarα for SUMO-dependent degradation (*Erker et al., 2013*), did show similarity to San1. In contrast to Rnf4/Snurf, Rn111 has also been shown to promote non-proteolytic, K63-linked ubiquitination of SUMO-modified substrate proteins (*Poulsen et al., 2013*).

In addition to Syvn1/Hrd1, Doa10 and Slx8 described above, also Hecw1 and Ufd4 have been connected with protein quality control. Specifically, Hecw1, also known as NEDL1, was found to ubiquitinate amyotrophic lateral sclerosis (ALS)-linked SOD1 variants, but not wild type SOD1 (*Miyazaki et al., 2004*). However, it is not clear that Hecw1 catalyzed SOD1 ubiquitination leads to proteasomal degradation. Genetic experiments have also linked the yeast E3 Ufd4 to protein quality control (*Theodoraki et al., 2012*), and showed that Ufd4 as a proteasome-associated E3 (*Xie & Varshavsky, 2002*) targets an orphan ER-membrane protein for degradation (*Ravid & Hochstrasser, 2007*).

## DISCUSSION

The recognition and degradation of misfolded proteins by the ubiquitin-proteasome system is of fundamental cellular importance. Only recently has it become clear that substrate recognition does not always rely on molecular chaperones, but can also occur through a direct binding mechanism involving intrinsic disorder, as employed by San1. We reasoned that this mechanism might well be employed by other E3 ligases, featuring sequence characteristics similar to San1 itself. While San1 is exclusively located in the nucleus (*Gardner, Nelson & Gottschling, 2005*), the substrate recognition mechanism employed by this protein could in principle also be employed by E3 ligases of the cytosol. Still, most of the proteins we identify as being San1-like have been shown to localize to the nucleus (*Her & Chung, 2009*; *Liu et al., 2006*; *Xu et al., 2009*). The underlying reason for this apparent bias is presently unknown, although nuclear proteins in general appear to be enriched for disorder as compared to, for example, cytosolic proteins (*Ward et al., 2004*). In addition, the nucleus appears to be a particularly active organelle in regard to protein quality control (*Nielsen et al., 2014*; *Park et al., 2013*).

In the case of San1, it has been shown that one reason for its unusually low content of lysine residues is to protect it from auto-ubiquitination and untimely proteasomal degradation (*Fredrickson et al., 2013*). Many of the disordered San1-like E3s that we identify here also display long stretches devoid of lysine residues. For instance, Rnf6 and Rnf12 both contain a remarkably long stretch of around 500 residues devoid of lysines, a feature conserved among their orthologs (Fig. 4). It is therefore possible that these proteins employ the same strategy as San1 to avoid auto-ubiquitination and degradation. Indeed, recent work shows a clear propensity for ubiquitinated lysines in proteasomal substrates to occur in regions of intrinsic disorder, and moreover these lysines are often located in close proximity to segments of >19 consecutive disordered residues (*Guharoy et al., 2016*). However, since lysine residues can undergo numerous post-translational modifications, including for instance sumoylation and acetylation, the low lysine content need not necessarily be connected with ubiquitination and degradation. Indeed, for Dma2, auto-ubiquitination followed by degradation was shown to occur despite the presence of a region with low lysine content (*Loring et al., 2008*).

It is noteworthy that both Rn111 and Slx8 belong to the rather small family of STUbL-type E3s. Since data suggest that several proteins are conjugated to SUMO in response to heat shock (*Tammsalu et al., 2014*; *Tatham et al., 2011*), such STUbL E3s are likely to be important for protein quality control. Indeed, like San1, Slx8 has been linked to protein quality control (*Wang, Jones & Prelich, 2006*). However, since the SUMO interacting motifs that are found in STUbLs are generally located in intrinsically disordered regions (*Vogt & Hofmann, 2012*), it cannot be ruled out that our approach is biased to detect STUbLs.

The concept that disorder can function to target protein misfolding in protein quality control is not only known from the case of San1. Recently, a disordered region of the cytosolic Bag6 protein, which functions to maintain misfolded aggregation-prone ERAD substrates in solution en route to the proteasome (*Wang et al., 2011*), has been demonstrated to bind and prevent aggregation of unfolded luciferase *in vitro* (*Xu et al., 2013*). Moreover, it has recently become clear that Bag6 harbors multiple "hydrophobicity recognizing" modules for binding of various substrates (*Tanaka et al., 2016*). Notably, when applying our procedure to the human Bag6 sequence we find a San1-similarity score of 0.51 (Fig. S2), similar to the highest scoring E3s in Fig. 5 (see Table S1). This suggests that our approach could be useful also for identifying other components of the cellular degradation machinery. In the case of Bag6, we speculate that the low lysine content present in the protein, although it is not itself an E3 ligase, protects the protein from ubiquitination by E3 ligases, such as RNF126, with which Bag6 directly interacts (*Rodrigo-Brenni, Gutierrez & Hegde, 2014*). Another recent example shows that a disordered C-terminal portion of the Ube2w E2 enzyme recognizes only intrinsically disordered N-termini of proteins to catalyze their N-terminal ubiquitination (*Vittal et al., 2015*). Indeed, we speculate that the San1-like proteins identified here might fall into two different classes. One class, which includes San1, Slx8 and Syvn1/Hrd1, appears to utilize its intrinsic disorder to identify a broader range of misfolded substrates, whereas the other class might rather use disordered regions to help recognize other substrates that perhaps themselves contain disordered regions. The identification here of a number of San1-like human E3 ligases suggests that

further examples could be added in the near future, opening up for more detailed functional and biophysical studies of the mechanisms by which intrinsic disorder in E3 enzymes is used to regulate protein homeostasis.

## METHODS

### HMM sequence-based search procedure

The hidden Markov model based sequence searches were conducted using the HMMER software package (http://hmmer.org/) by constructing a model using the jackhmmer tool, and subsequently using the hmmsearch program to search through the entire Uniprot databases of *Saccharomyces cerevisiae* and human, respectively. Default values were used for these programs. The top-10 matches for human and top-5 hits for yeast are included as Fig. S1.

### Datasets

For the remaining analyses, we searched directly within a set of 80 yeast E3 ligases (*Li et al., 2008*) and a set of 563 human E3 ligases recently published by Tompa and coworkers (*Bhowmick et al., 2013*). This set was constructed as a merge of an extract from the KEGG BRITE database (*Kanehisa & Goto, 2000*) and well-characterized E3 ligases from a previously published dataset (*Li et al., 2008*), filtered for uncertain annotations, identical gene names, and internal sequence homology (>85%), leading to an unbiased, representative set of the currently known E3 proteins. It should be noted, however, that additional human E3 ligases probably exist (*Bhowmick et al., 2013*; *Li et al., 2008*), and more than 600 RING-type E3 ligases are likely expressed in human cells (*Deshaies & Joazeiro, 2009*).

All results obtained in the analysis for this paper are made available as an annotation of this original data set, presented as an Excel file in Table S1. This dataset contains all position-specific prediction results, it provides the resulting San1 similarity score, and allows for easy sorting of candidates based on the criteria outlined in the paper. The spreadsheet also permits reanalyses of the data with different parameter values (e.g., the lengths of the sliding window and the ANCHOR stretch).

### Prediction of protein binding regions

We used the ANCHOR (*Meszaros, Simon & Dosztanyi, 2009*) program to assign position-specific scores for protein binding within disordered regions. For the purpose of the San1 similarity score, we considered stretches of between 15 and 40 residues (see below) for which the ANCHOR scores are all higher than 0.5. The total number of residues occurring within such regions was normalized by the length of the window to give an ANCHOR stretch score. No requirements were imposed on the disorder status of such regions.

### Choice of parameters

The size of the sliding window and the boundaries defining an ANCHOR stretch (15–40), are free parameters that we chose based on the observed features of San1. The window size for lysine content was set to 150, which is compatible with the particular stretch found in San1. Choosing larger values penalizes smaller proteins (candidates that are smaller

than the window size will be assigned a score of zero), and choosing smaller values makes the score less sensitive to long lysine-free stretches. The 15 and 40 residue ANCHOR stretch boundaries were based directly on the lengths observed in San1 (*Rosenbaum et al., 2011*). We probed the sensitivity of our approach to these choices by considering the top candidates with varying values for both of these parameters and found the presented results to be quite robust, as long as the chosen values were not directly at odds with the characteristics observed for San1. For instance, when changing the ANCHOR stretch length to 5–35, we find that 17 out of 20 of the Top 20 candidates in Table S1 are preserved.

### Funding

This work was supported by grants from the Lundbeck Foundation (SVN, RHP: R108-A10121; LE: R83-A7446), Danish Council for Independent Research/Natural Sciences (RHP: 18490), the Villum Foundation (WB: VKR023445), the Novo Nordisk Foundation (KLL, RHP) and the Danish Cancer Society (RHP, KLL). The funders had no role in study design, data collection and analysis, decision to publish, or preparation of the manuscript.

### Grant Disclosures

The following grant information was disclosed by the authors:
Lundbeck Foundation: R108-A10121, R83-A7446.
Danish Council for Independent Research/Natural Sciences: 18490.
Villum Foundation: VKR023445.
Novo Nordisk Foundation.
Danish Cancer Society.

### Competing Interests

The authors declare there are no competing interests.

### Author Contributions

- Wouter Boomsma conceived and designed the experiments, performed the experiments, analyzed the data, wrote the paper, prepared figures and/or tables, reviewed drafts of the paper.
- Sofie V. Nielsen performed the experiments, reviewed drafts of the paper.
- Kresten Lindorff-Larsen and Rasmus Hartmann-Petersen conceived and designed the experiments, analyzed the data, wrote the paper, reviewed drafts of the paper.
- Lars Ellgaard conceived and designed the experiments, analyzed the data, wrote the paper, prepared figures and/or tables, reviewed drafts of the paper.

### Data Availability

Github: http://github.com/e3-disorder/e3_table/.

## Supplemental Information

Supplemental information for this article can be found online at http://dx.doi.org/10.7717/peerj.1725#supplemental-information.

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
