# Peer review of "Bioinformatics analysis identifies several intrinsically disordered human E3 ubiquitin-protein ligases"

_PeerJ, doi:10.7717/peerj.1725_

## Round 0.1 · original submission · Major Revisions

Please address critical points raised by two reviewers and revise manuscript accordingly.

Reviewer 1 ·

Basic reporting

No Comments

Experimental design

A few suggestions:
1. You could try the new alignment method (http://www.ncbi.nlm.nih.gov/pubmed/26568635) that was designed specifically for the disordered proteins/regions to enrich your search for the San1 orthologs.
2. You could try D2P2 or MobiDB databases to collect a more comprehensive putative annotation of disorder, which perhaps would provide a more reliable prediction.
3. While ANCHOR predicts disordered protein binding regions, you should also consider annotating MoRF regions. There are a few recent methods that you could use including MoRFpred, MoRFChibi, and DISOPRED3.

Validity of the findings

The findings are computational in nature and the authors appropriately discuss them as putative. The hypothesis derived based on these results is supported by a logical discussion. One of the strengths of this article is the new scoring function and the authors provide a convincing quantitative argument to support its validity.

Additional comments

The article is interesting and innovative, particularly in the sense of developing a new approach to find orthologous proteins in the presence of intrinsic disorder.

Reviewer 2 ·

Basic reporting

Overall the article is clear and well-written. A minor point is that the PeerJ guidance suggests separate results and discussion sections. In my personal opinion, the combined results and discussion are not a detriment to this article, but I am unsure if this is a hard requirement of the journal. On a related note, in a number of places in the text (e.g. line 213) the reader is referred to the discussion section. I presume these should be changed to the conclusion section.

The protein names used throughout the manuscript should be checked carefully. For example, line 183 refers to protein RNF111, whilst table 1 contains RN111. Similarly, throughout the text the name HRD1 is predominantly used, however this is referred to as SYNV1 in table 1. In addition, adding the Uniprot reference IDs to table1 would add clarity, especially given that the S. cerevisiae strain is not identified.

Figure 1 does not explain that the black dots represent Lysine residues, so the figure legend should be updated. This is also referred to on lne 147. In all figures, it was not clear to me if each dot represented a single lysine residue.

In the legend for Figure 4, it is noted that the score is "not defined for the first and last positions". It would be cleared for the reader to be explicit that this is, presumably, the first and last 125 residues.

Otherwise all figures and tables are well presented.

Experimental design

My main concern centres over the identification of San1-like proteins. The authors note that PSI-BLAST/DELTA-BLAST do not detect any obvious human candidates. However, it is well established that HMM-based methods are more sensitive and thus better able to reveal remote homology. This extra sensitivity appears to be crucial in this particular case. Applying standard practice I was able to quickly generate a list of potential human candidates.

For reference, my workflow was to build an HMM using jackhmmer from the HMMER package, starting with the sequence San1/P22470 and searching against Uniprot/Trembl. I then took this HMM and searched the human sequences from Ensembl with the hmmsearch program. This process reveals a number of potential candidates that have homology to P22470 both inside and outside of the RING domain. These include some of the top-scoring proteins identified by the authors in table1, as well as many others. In case it is useful, the HMM built can be found here (http://pastebin.com/Gn0pziEb) and the results (--domtblout) of the search can be found here (http://pastebin.com/xwDqUh2X).

I would suggest that this approach to homology search is a more rigorous method and should be strongly favoured over the non-standard method used here. As well as being an established method, HMM methods are statistically robust and have been extensively benchmarked.

Finally, there are a number of specific points to raise with respect to the developed method. To be clear, these are independent of the more suitable approach outlined above and addressing these would not alter my opinion that an HMM method can and therefore should be used.
1.) It is not entirely clear if the San1 similarity score does indeed work. Calculating the score for all human proteins should ideally reveal that non-E3 ligases receive a much lower score.
2.) Calculating a consensus disorder prediction by a residue-wise average of the the individual disorder predictors should be justified. Other approaches to consensus prediction (e.g. as available in the MobiDB and D2P2 databases) apply a voting system which protects against any one predictor outputting particularly high or low raw values.
3.) Similarly, a justification for using the maximum score for any residue as the protein's san1 similarity score would be useful.
4.) It was not entirely clear to me how the the ANCHOR component of the score was calculated. Did this score require regions anywhere between 15 and 40 residues in length that all had ANCOR scores > 0.5? Were these regions additionally constrained such that all residues had a (consensus) disorder prediction > 0.5?
5.) It was not obvious why the specific proteins were chosen in lines 182-184. These are not simply the highest scoring sequences, so justification for this is important.

Validity of the findings

No further comments

Additional comments

That a number of ligases (in addition to San1) may directly recognise their targets using the flexibility allowed by intrinsically disordered regions is interesting. I feel that using a standard method and revisiting the analysis of results could yield a nice piece of work.

Reviewer 3 ·

Basic reporting

No comments

Experimental design

No comments

Validity of the findings

No comments

---

## Round 0.2 · accepted · Accept

Thank you for serious consideration of the reviewers' comments and for the rigorous revision of the manuscript.